# Optimal Reconstruction of Single-Pixel Images through Feature Feedback Mechanism and Attention

Zijun Gao *, Jingwen Su, Junjie Zhang *, Zhankui Song, Bo Li and Jue Wang

School of Information Science and Engineering, Dalian Polytechnic University, Dalian 116034, China; songzhankuiwudi@163.com (Z.S.); libolb@dlpu.edu.cn (B.L.); wangjue@dlpu.edu.cn (J.W.)
* Correspondence: gaozj@dlpu.edu.cn (Z.G.); zjj98818@126.com (J.Z.)

**Abstract:** The single-pixel imaging technique can reconstruct high-quality images using only a bucket detector with no spatial resolution, and the image quality is degraded in order to meet the demands of real-time applications. According to some studies of algorithm performance, the network model performs differently in simulated and real-world experiments. We propose an end-to-end neural network capable of reconstructing 2D images from experimentally obtained 1D signals optimally. In order to improve the image quality of real-time single-pixel imaging, we built a feedback module in the hidden layer of the recurrent neural network to implement feature feedback. The feedback module fuses high-level features of undersampled images with low-level features through dense jump connections and multi-scale balanced attention modules to gradually optimize the feature extraction process and reconstruct high-quality images. In addition, we introduce a learning strategy that combines mean loss with frequency domain loss to improve the network's ability to reconstruct complex undersampled images. In this paper, the factors that lead to the degradation of single-pixel imaging are analyzed, and a network degradation model suitable for physical imaging systems is designed. The experiment results indicate that the reconstructed images utilizing the proposed method have better quality metrics and visual effects than the excellent methods in the field of single-pixel imaging.

**Keywords:** single-pixel imaging; feature feedback; feature extraction; learning strategy

## 1. Introduction

In a single-pixel imaging (SPI) system [1,2], the reference beam is structurally modulated to illuminate the target scene. For measurement, the single-pixel detector samples the reflected light several times from the target scene. Based on the second-order correlation of the fluctuations in the light field, spatial information about the target can be calculated [3]. In low-light conditions, single-pixel detectors with larger active areas are easier to fabricate and have higher light sensitivity. It is possible to build a low-cost imaging system using SPI in low-light environments, making it widely used in optical imaging, including Spectral Image Analysis [4], 3D Imaging [5,6], Remote Sensing [7], and Terahertz Imaging [8–10].

The SPI light encodes some of the target scenes corresponding to multi-row coordinates and column coordinates and acquires one-dimensional measurements by a pre-built imaging system for the encoded light. Sun et al. reconstructed images with a resolution of $256 \times 256$ using 106 random matrices (the number of measurements is much larger than the number of pixels in the reconstructed image) [11]. However, the quality of the reconstructed images is extremely poor and differs greatly from conventional imaging systems based on a surface array sensor [12]. Due to its extremely long acquisition time and poor quality of image reconstruction, SPI is not suitable for all applications [13–16]. The deterministic model-based measurement strategy is an effective method of preventing long acquisition times for SPI and low-quality reconstructed images [17]. Hadamard Single Pixel Imaging is a method that uses a part of the Hadamard matrix as the measurement matrix [18].

HSPI can acquire the Hadamard frequency domain and reconstruct the target scene using inverse operations. HSPI requires fewer measurements for accurate reconstruction than, for example, Fourier Single Pixel Imaging (FSPI) [19]. In terms of image reconstruction, HSPI is more efficient and provides superior results. Several studies have shown that by optimizing the ordering of Hadamard bases, memory consumption for matrix generation and storage can be reduced, and image reconstruction speed can be increased [20,21]. Sun et al. have designed the Russian Dolls sort to simplify image reconstruction at a later stage [22]. Yu proposed a new cake-cutting method that achieves an optimized pattern sequence by directly rearranging the number of internal slices of the base pattern in ascending order [23]. In some cases, this enables a more accurate reconstruction of images under conditions of low noise. However, it is still difficult for some real-time applications to strike a balance between the number of measurements and the quality of the image.

Deep learning (DL) is widely used in the image field for its excellent feature extraction capabilities [24,25]. In a recent SPI study, Higham et al. developed a Deep Convolutional Auto-encoder Network (DCAN) and degradation model to reconstruct undersampled single-pixel images in real time by training the degraded signal against a clean signal [26]. Saad et al. proposed a framework that combines Skip-connection and denoising self-coding structures [27], in which Skip-connection preserves image features and then recomputes them in order to reconstruct a high signal-to-noise image using predetermined information. Lu et al. combined Generative Adversarial Networks (GAN) with U-net networks to propose the SPI-CGAN model [28], which uses Wasserstein loss to reconstruct images at low sampling rates. However, reconstructing high-quality images with low sampling rate signals is still challenging [29]. Since the low sampling rate signal contains a limited amount of scene information, the boundary between the reconstructed image subject and the background can be blurry, and the texture information can be difficult to reconstruct. Recent research has applied feedback mechanisms to network architectures to refine high-level features and return them to the previous convolutional layers [30]. It is possible to refine low-level coded information through top-down working, which can improve the network's ability to learn the features of an image as a result. Currently, deep learning-based SPI tasks focus solely on transforming low-level features into high-level ones, neglecting to map and transfer high-level features into low-level ones. Additionally, existing deep learning-based SPI methods generally rely on a single sampling rate and a fixed value of noise as the primary source of image degradation [31]. As the noise level of the simulated degraded image cannot be matched to that of the real undersampled image, significant differences in performance are observed between the training model in simulated experiments and in real-world applications [32].

To address the above problems, this paper proposes a new SPI network, the Single-Pixel Imaging Feedback Attention Network (SPIFAN), which utilizes high-level features to guide (refine) undersampled low-level features through feedback connections. The proposed SPIFAN consists of an RNN with a Feedback Block (FB), which is composed of a dense residual block and an attention block. The dense residual block is capable of capturing rich high-level features from undersampled image features and associating them with low-level features. The attention module is composed of a dual attention structure consisting of a channel attention module and a spatial attention module, resulting in a more informative representation of sampled signal features. As the feature weights of the SPI sampled signals differ in position, we utilize a hybrid dilation Conv layer with different dilation rates during feature extraction to obtain a larger perceptual field, balancing the effect of feature weights on the overall detail of the reconstructed image, resulting in better image reconstruction. The proposed method utilizes the hidden state in each iteration as part of the input for the next iteration, ensuring that the input for the next iteration contains high-level features of the image. A combination of image mean loss and frequency loss is used to help the network progressively learn complex single-pixel undersampled images. In addition, a complex and practical degradation model for SPI is designed in order to approximate the real HSPI undersampled images to the extent possible, which

consists of a random low-pass filter (1–10%) and multiple noise degradations. Simulation and experimental results show that using our proposed method improves the sharpness and detail texture of undersampled images while avoiding artifacts and blurring. The reconstructed images have the best visual effect compared to the existing excellent SPI image reconstruction methods. In terms of quality metrics, the reconstructed images have the best numerical performance. The proposed method compares favorably with traditional HSPI methods and advanced SPI methods.

## 2. Related Work

In this section, we present work related to SPI based on deep learning. First, we describe HSPI and discuss the balance between the sampling rate and the quality of real-time images. Then, we discuss what causes SPI image degradation. Finally, we discuss the shortcomings of the current approach to SPI that uses DL.

### 2.1. HSPI

Figure 1 illustrates the SPIFAN imaging system used in this paper. The HSPI reconstructs the target image by acquiring the Hadamard spectrum of the target scene and using the inverse Hadamard Transform. The Hadamard spectrum consists of a set of Hadamard coefficients, each of which corresponds to a unique Hadamard basis. The lens is used to form an image of the Object on the digital micromirror device (DMD), using a DMD to realize a number of projections of a reference beam modulated by the corresponding Hadamard basis. A photomultiplier tube (PMT) measures the intensity of the reflected light and digitizes the signal via the Digitizer. Second-order correlations for image reconstruction are computed using the Computer based on optical field fluctuations. The proposed SPIFAN algorithm achieves high-quality image reconstruction.

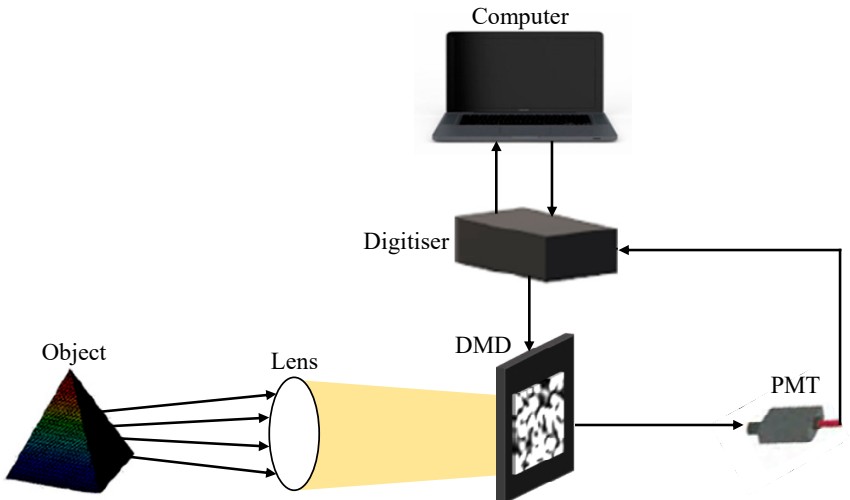

**Figure 1.** Diagram of the experimental setup of the SPIFAN system.

The single-pixel light intensity measurement is mathematically equivalent to the inner product of the Hadamard basis and the object. The two-dimensional Hadamard transform $H\{\}$ with the target image $I(x, y)$ is defined as:

$$\widetilde{I}_H(u,v) = H\{I(x,y)\} = \sum_{x=0}^{M-1}\sum_{y=0}^{N-1} I(x,y)(-1)^{q(x,y,u,v)} \tag{1}$$

where $(x, y)$ are the coordinates of the spatial domain of $I(u, v)$ are the coordinates of the Hadamard domain of $I$, and $\widetilde{I}$ is the results of the inverse Hadamard transform of image

I. $M$ and $N$ represent the rows and columns of the Hadamard matrix, respectively, where $n = log2N$ and:

$$q(x, y, u, v) \equiv \sum_{i=0}^{n-1} [g_i(u)x_i + g_i(v)y_i] \tag{2}$$

$$\begin{aligned} g_0(u) &\equiv u_{n-1} \\ g_1(u) &\equiv u_{n-1} + u_{n-2} \\ g_2(u) &\equiv u_{n-2} + u_{n-3} \\ &\vdots \\ g_{n-1}(u) &\equiv u_1 + u_0 \end{aligned} \tag{3}$$

the terms $u_i$, $v_i$, $x_i$, and $y_i$ are the binary representations of $u$, $v$, $x$, and $y$, respectively.

In some practical applications, in order to increase the imaging rate, it is common practice to decrease the sampling rate and reduce the number of measurements made on the object, thereby reducing the total imaging time. The sampling rate is usually controlled at 1–10%, which is the cause of blurring and distortion in the reconstruction, greatly limiting the application scenarios of SPI.

### 2.2. Feedback Mechanism

Feedback is defined as the output of the system as part of the input during causal iteration, which in turn influences the work of the system [33–36]. In DL, the feedback mechanism can support the network in using high-level features of the output to optimize the weights of the previous convolution kernel. Recent research has shown that feedback mechanisms have been applied to a variety of computer vision tasks [37–40]. For the SPI task, Antonio et al. used an RNN network with temporal memory to control the delivery and loss of features [41]. Ikuo et al. improved the quality of the reconstructed images by chunking the measurement data input and accumulating and updating them in an RNN [42]. These methods propagate feature information in a way that includes only the forward propagation of low-level features to high-level features in the process of going from an undersampled image to a final high-quality image, ignoring the role of high-level features in guiding low-level features. We propose a Feedback Block (FB) in this paper as the basis of SPIFAN in order to maximize the use of high-level features. In the FB, feature information is propagated between levels through dense skip connections in order to optimize feature mapping and extraction.

### 2.3. Attention Mechanism

The observation matrices used for compression perception are generally orthogonal in nature and have the property of energy concentration. The more uniformly distributed the image data is, the more the data in the resulting simulated compressed sample value matrix is concentrated on the edges. Essentially, this mechanism highlights important characteristics of the input object and reassigns feature weights accordingly.

Zhang proposed the Residual Channel Attention Networks (RCAN) based on the channel attention mechanism [43]. Hu et al. proposed the Squeeze-and-Excitation module to obtain channel attention by globally averaging the pooling of input features. The spatial attention mechanism focuses on which information features are important in the spatial domain [44], with a high weight indicating a high importance to that frequency domain. Through weighting operations, the model can effectively focus on relevant features and ignore irrelevant features. Wang et al. proposed a residual attention network where the attention module is designed as a codec structure [45]. Woo S et al. proposed CBAM (Convolutional Block Attention Module), which is a hybrid attention module based on spatial attention and channel attention [46].

### 2.4. Degradation Models

In a physical SPI system, the measurements are corrupted by a mixture of Gaussian and Poisson noise [47]. Gaussian noise is independent of the measurement signal and is generated by fluctuations in the circuit; Poisson noise is related to the measurement signal and is generated by the discrete nature of the charge. The instability of real noise can lead to different levels of distortion in the reconstructed image [48]. Therefore, when the training model in its ideal state is used in a physical SPI system, its performance can be severely degraded.

## 3. Materials and Methods

In this section, we describe the SPIFAN architecture in detail. We begin by introducing the network as a whole and introducing the feedback mechanism. Following this, we describe in detail the dense residual block as well as the attention module of the feedback module. To obtain more accurate training data, we design a suitable degradation model for undersampled signal features.

### 3.1. Overview of Network Architecture

The single-pixel image reconstruction feedback network proposed in this paper consists of three parts: feature extraction, feedback, and image reconstruction. Figure 2 illustrates the network structure.

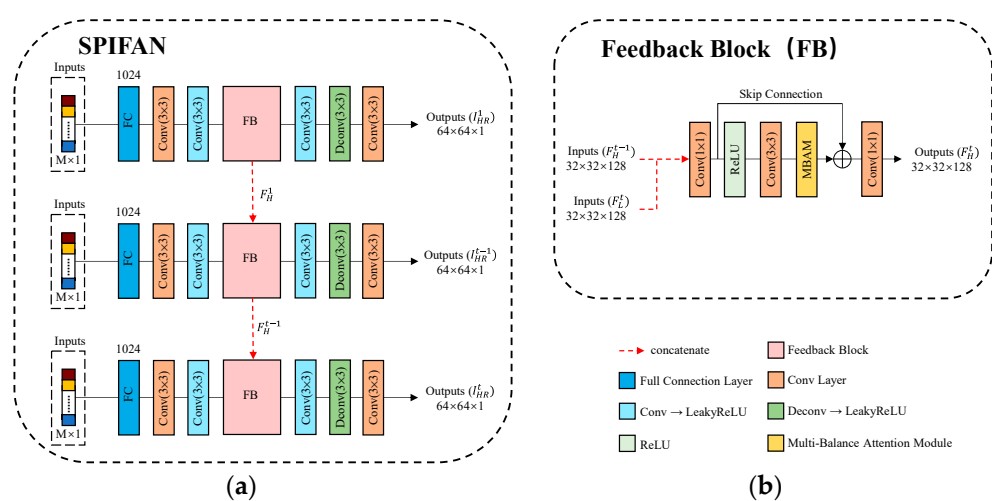

**Figure 2.** (**a**) Overall network structure of SPIFAN; (**b**) structure of the feedback module in SPIFAN.

Our feedback network uses simulated 1D bucket signals as input to perform the reconstruction of high-quality images directly from 1D undersampled signals. Firstly, a fully connected layer of size 1024 by 1 is used to determine the internal correlation of the undersampled signal, and next, a powerful single-pixel feedback reconstruction network is utilized to reconstruct the target image. In the feature extraction part, two Conv layers $(3 \times 3)$ are utilized to achieve dimensionality reduction of the low-level features in the stream normalized by the fully connected layer and are used as part of the input to the feedback module.

As part of the first iteration, $F_L^1$ is assumed to be the high-level feature $F_H^0$ of the FB output at the same time as the initial FB input. For the t-th iteration, $F_L^t$ and $F_H^{t-1}$ are spliced together and used as inputs to the FB. $F_H^{t-1}$ can be used as feedback information to guide the extraction of low-level features. The output of FB, $F_H^t$, is used both for image reconstruction in $I_{HR}^t$ and to guide low-level feature extraction in the next iteration. In the image reconstruction part, the high-level features are upsampled by two deconvolution layers to recover the feature map size to $64 \times 64$. Reconstruction of the final image is completed by the Conv layer $(1 \times 1)$.

### 3.2. Feedback Block

The feedback module uses the output high-level features to refine the low-level features, thus enabling optimization of the feature extraction process for the next iteration. The FB contains a dense residual module and a balanced attention module, and the connections between the various groups are made using dense jump connections. This enhances the network's capacity, reduces gradient disappearance, optimizes high-level features, and is useful for reusing features. In addition, each mapping group does not contain a Batch Normalization layer (BN layer). For training, the BN layer takes the mean and variance of the batch, and for testing, it takes the mean and variance of the whole dataset. In situations where the training and test sets differ significantly, the BN layer may negatively affect the visual impact of the reconstruction and minimize the generalizability of the model. It has been shown that, in some PSNR-oriented image reconstruction tasks (such as Super-resolution [49], Deblurring [50], etc.), eliminating the BN layer improves network performance and generalization and simplifies computation.

At the $t$-th iteration, FB outputs the more accurate high-level feature $F_H{}^t$ after combining the high-level feature $F_H{}^{t-1}$ from the previous iteration with the low-level feature $F_L{}^t$ from the current iteration. In FB, the low-level feature $F_L{}^t$ and the feedback information $F_H{}^{t-1}$ are concatenated and compressed by a $1 \times 1$ Conv layer to produce the first low-level feature $L_0{}^t$. After feature dimensionality reduction is the dense residual module (RD), which consists of a dense residual unit consisting of a $3 \times 3$ convolution with ReLU activation and a $1 \times 1$ convolution layer. This part computes the input fusion features to fully learn and extract the deep features. The output of the last dense residual unit is connected to the input of the RD module, which performs feature fusion and dimensionality reduction by $1 \times 1$ convolution and serves as the input to the Balanced Attention Module (BAM). The BAM contains two paths for channel and spatial attention. In the channel attention path, the input $C \times H \times W$ feature maps are transformed into $C \times 1 \times 1$ channel-level feature maps by averaging the pooling layers. Then, feature dimensionality reduction and activation operations are implemented by two $1 \times 1$ Conv layers and a ReLU activation layer. The feature map is then transformed into a $C \times 1 \times 1$ output weight vector using a Sigmoid layer. In the spatial attention path, the input $C \times H \times W$ feature map is converted into a spatial feature map of size $1 \times H \times W$ by the maximum pooling layer. The $7 \times 7$ Conv layers and Sigmoid layers then convert the feature maps into spatial output weight vectors of size $1 \times H \times W$. In order to obtain the weight values of the attention features, the two output weight vectors are dot produced with the input feature map. Finally, the feedback feature $F_H{}^t$ is output through the $3 \times 3$ Conv layer. The output $F_H{}^t$ serves two purposes: on the one hand, it is used as input to the feedback module in the next iteration to improve the efficiency of the feature mapping in refining low-level features. On the other hand, as a high-level feature, it assists the SPIFAN network in completing the image reconstruction for this iteration.

### 3.3. Multi-Balance Attention Module (MBAM)

The use of large convolutional kernels in SPI networks based on attention mechanisms enables a more compressed undersampling of the output feature map. However, the use of large convolutional kernels is limited in the feature extraction part because of the large number of computational parameters they add to the network. Therefore, we propose a multi-scale balanced attention module, MBAM, which captures the multi-scale patterns of upsampled feature maps using dilation convolution. MBAM consists mainly of dilation convolution with a convolution kernel size of 3 and a dilation rate of [1,2,5]. Figure 3 shows the structure of MBAM.

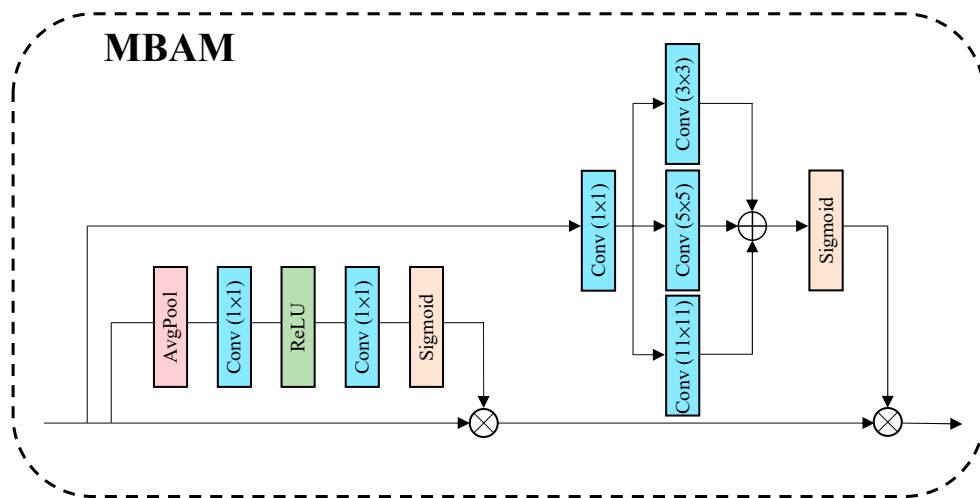

**Figure 3.** MBAM structure diagram.

The equivalent convolution kernel size for the dilated convolution shown in Figure 3 is calculated as follows:

$$K = k + (k-1)(n-1) \tag{4}$$

where $K$ denotes the equivalent convolutional kernel size, $k$ denotes the actual convolutional kernel size, and n denotes the expansion rate. Therefore, the equivalent kernel sizes for the dilated convolution with kernel size 3 and dilation rate [1, 2, 5] are 3, 5, and 11, respectively. By using dilated convolution, a larger perceptual field can be obtained without requiring higher network parameters, allowing the image reconstruction network to improve the mapping between compressed signal features and optimally reconstructed images without increasing feature extraction computations. Additionally, the dilation convolution with different dilation rates can capture multi-scale information in the feature map, allowing the network to recognize the significance of each part of the sampled signal for the purpose of optimal image reconstruction. The output of MBAM is calculated as follows:

$$\mathbf{Y} = \mathbf{X} + MBAM(MDC(\mathbf{X})) \tag{5}$$

where $\mathbf{X}$ denotes the input feature map, $\mathbf{Y}$ denotes the output feature map, $MBAM$ denotes the Multi-scale Balanced Attention Module, and $MDC$ denotes Multi-scale Dilated Convolution.

*3.4. Degradation Module*

Signal compression is a degradation unique to SPI systems. The more the signal is compressed, the less information is available for 2D image reconstruction, which results in poorer-quality reconstructed images. In the reconstruction network where PMT measurement values are mapped to 2D images, obtaining more and more realistic PMT simulated measurement values for network training can significantly improve the quality of reconstructed images.

Each 2D image in the original dataset had been resized to 64 × 64, converted to greyscale, and histogram equalized to create a single-channel 2D image. The single-channel image is then normalized by inner-producing it with the Hadamard observation matrix to give the equivalent of a fully sampled photometric value. In other words, the 4096 pixels of each image in the dataset itself are linearly transformed into Hadamard coding space. Since the high-frequency image information in Hadamard's coding space is concentrated in a small fixed area, it is possible to simulate compressive sampling of the SPI more easily. In the end, the compressed sampling of the image signal is simulated by thresholding the photometric values. As the Hadamard mode contains positive and negative values, a differentiation method is used to obtain the simulated sampled signal. It is assumed that m$\alpha$+ denotes a positive measurement of Hadamard mode and m$\alpha$- denotes a negative

measurement of Hadamard mode. The noise is a composition of Poisson and Gaussian distributions. Gaussian noise in the SPI system is generated by fluctuations in the circuit and is uncorrelated with the channel, generating all noise and errors consisting of {0, 1} [51]. The Poisson noise in the SPI system is generated by the discrete nature of the charge, is signal-dependent, and satisfies the Poisson distribution. The Hadamard simulated sampling values are assumed to be:

$$\hat{m}_{+,-}^{\alpha} = KP(\alpha H_1^{+,-} f) + N(\mu_{dark}, \sigma_{dark}^2) \tag{6}$$

where $P$ and $N$ are Poisson and Gaussian distributions, $K$ is a constant representing the overall system gain, $\alpha$ is the intensity of the image in photon photons (proportional to the integration time, which is proportional to the integration time), $\mu dark$ is the dark current (counts), and $\sigma dark$ is the dark noise (counts). Further, we assume that the magnitude of $\mu dark$ and $\sigma dark$ does not depend on the intensity of the image, which results in the following normalized measure:

$$m^{\alpha} = (m_+^{\alpha} - m_-^{\alpha})/(\alpha K) \tag{7}$$

*3.5. Loss Function*

In this paper, we propose to optimize the proposed network based on a combination of $L_1$ Loss ($L_1$) and Hadamard Frequency Loss ($L_H$). $L_1$ and $L_H$ are defined as, respectively:

$$L_1 = \frac{1}{w \cdot h} \sum_{x,y} |I_{HR}(x,y) - I(x,y)| \tag{8}$$

$$L_H = \frac{1}{2} \|H(I_{HR}) - H(I)\|_2^2 \tag{9}$$

where $w$ and $h$ denote the width and height of the image, respectively, $I_{HR}$ is the output image of the generator, $I$ is the original image, and $H()$ corresponds to the Hadamard frequency domain transform. The combined loss of $L_1$ and $L_H$ is:

$$L = a \cdot L_1 + b \cdot L_H \tag{10}$$

Following the calculation of the loss functions separately, we set $a$ and $b$ to 1 and 0.1, respectively, depending on the magnitude of the results. The strategy orders ($I_{HR}^1$, $I_{HR}^2$, ..., $I_{HR}^t$) according to the difficulty of reconstructing the images and the loss calculation is completed by t iterations. The loss function in the network can be expressed as:

$$L(p) = \frac{1}{T} \sum_{t=1}^{T} W^t \|L\|_1 \tag{11}$$

where $p$ denotes the parameters of the network, and $W^t$ is a constant factor indicating the value of the output at the $t$-th iteration. In this paper, each reconstructed image is considered to contribute equally to the optimization network, and therefore, $W^t$ is set to 1.

## 4. Results

*4.1. Training and Validation*

4.1.1. Dataset

In this paper, the STL-10 [52] dataset is selected as the training dataset, which contains images of 96 × 96 size. This dataset consists of ten classes: monkeys, cats, dogs, deer, cars, trucks, planes, birds, horses, and boats. The SPIFAN network was trained on compressed signals using sampling rates of 2%, 5%, and 8%. A total of 10,000 unlabelled standard images were used for the training process. As part of the training process, a test set of 1000 images was used to validate the performance of the network, and a validation set (Set5 [53], Set14 [54], and Urban100 [55]) was used to evaluate the final model's performance.

During model training, these datasets are not visible. The proposed model is compared with conventional HSPI using the validation dataset at different sampling rates.

### 4.1.2. Implementation Details

In the training stage, the standard image is randomly cropped to $64 \times 64$ and fed into the degraded model to obtain a one-dimensional analog signal. Both the proposed method and the other methods use the Adam optimizer to optimize the model with $\beta_1 = 0.9$, $\beta_2 = 0.999$, and $\varepsilon = 1 \times 10^{-8}$. L2 regularisation is used to reduce overfitting. We set the model weight decay to $1 \times 10^{-8}$. In the training phase, Monte Carlo cross-validation was used to find the optimal hyperparameters (i.e., initial learning rate and number of iterations). The initial learning rate was set to $1 \times 10^{-4}$ and halved every 50 iterations. The proposed model was implemented using the PyTorch framework and trained on an NVIDIA GTX3090 GPU. Figure 4 shows a schematic of SPIFAN training and validation loss variation. It can be seen from Figure 4 that the training $L_1$ Loss and $L_H$ Loss decrease as the number of iterations increases. At around 200 iterations, the loss values stabilized. This indicates that the learning rate and the coefficients of the loss function are set appropriately. The trend of verifying the loss values with the number of iterations indicates that the SPIFAN network learns normally and no overfitting occurs.

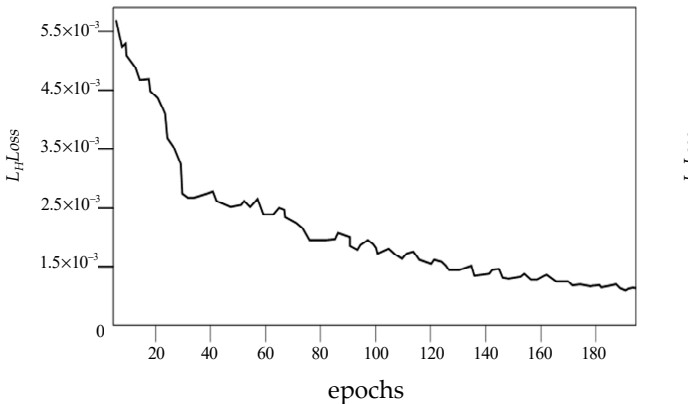 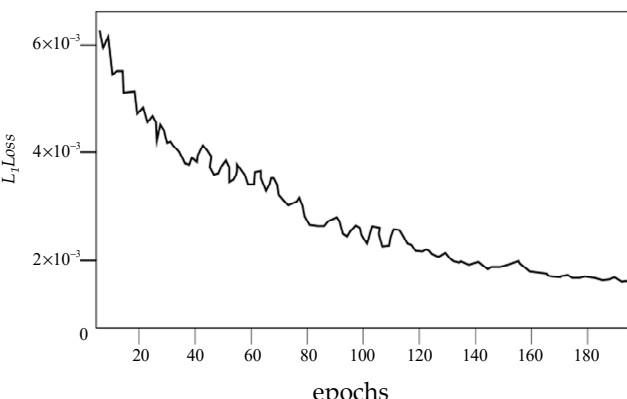

**Figure 4.** Graph of SPIFAN losses.

### 4.1.3. Evaluation Metrics

Peak Signal-to-Noise Ratio (PSNR) and Structure Similarity Index (SSIM) are commonly used metrics for evaluating the quality of reference images. The PSNR is used to reconstruct images from contrast scenes, and it indicates the degree of distortion by comparing the pixel gaps between the reconstructed and reference images. The SSIM is used to measure the degree to which the reconstructed image is similar to the original image. These metrics are calculated from a combination of Luminance, Contrast, and Structure, respectively.

### 4.2. Image Reconstruction Results

The proposed method was tested on different types of salient targets, including a bird, butterfly, zebra, pepper, baboon, and face, in order to verify its effectiveness. In order to observe the degradation of the undersampled reconstructed image, a test image (Bird) was selected to simulate the HSPI compression reconstruction, and the sampling rate was chosen as 1–50%. Figure 5 illustrates the reconstruction results.

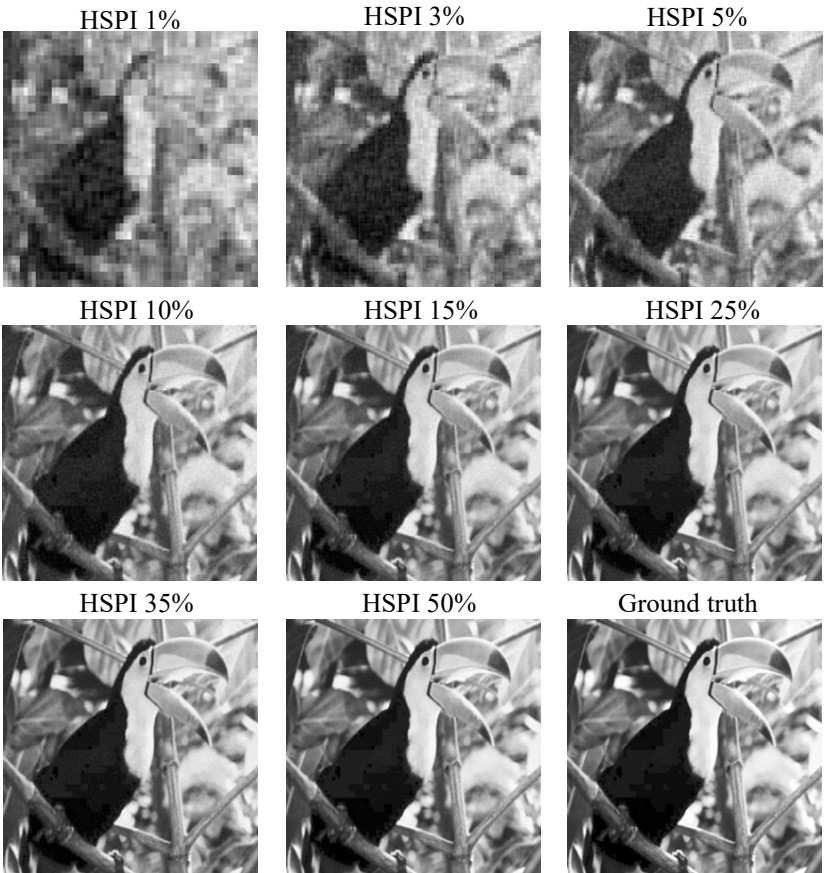

**Figure 5.** Partial reconstruction of Bird images from the test set by HSPI.

It can be seen in Figure 5 that the reconstructions with S between 1% and 10% are less clear and show a significant block blur in the noiseless condition. In comparison with other reconstructions, the reconstruction at S = 25% is clearer. Nevertheless, in some application scenarios involving real-time imaging, the reconstruction time of an HSPI image with a sampling rate of 25% is too long, and a lower sampling rate (S < 10%) is usually used. However, low-sampling images are too blurry, have significant noise, and are extremely poorly visualized. Therefore, it is necessary to apply the proposed method to generate high-quality reconstructed images from undersampled signals.

To verify whether the proposed method can help conventional HSPI to improve the quality and visual effect of the reconstructed images, the undersampled signals after the degradation model reconstruction were fed into SPIFAN for optimal reconstruction. Ground truth, HSPI reconstruction, and SPIFAN reconstruction were compared qualitatively and quantitatively with sampling rates S of 2%, 5%, and 8%, respectively, and peak signal-to-noise ratio (PSNR) and structural similarity (SSIM) were chosen as evaluation metrics for the images [56]. The Ground truth and reconstruction results are shown in Figure 6.

The proposed SPIFANs are all capable of obtaining clearer images of higher quality from undersampled signals compared to the corresponding sample rate HSPI reconstructions. Furthermore, there is a link between the sampling rate and the quality of the SPIFAN reconstruction. As the 2% undersampled signal contains too little feature information to be used for reconstruction, the network model is only able to capture coarse information about the target scene, resulting in limited visual enhancement. Therefore, the quality of the reconstructed image is further impacted by lower sampling rates. In contrast, as the sampling rate is increased, the network model is able to capture more accurate information about the target scene, and the reconstruction quality of the image is enhanced. In comparison to SPIFAN 2%, SPIFAN 5% shows a more pronounced improvement in terms of visual

results. Due to the highest sampling rate, the reconstruction result for SPIFAN 8% is also the best of all results. The mean values of the quality metrics for the reconstructed images at each sampling rate in Figure 5 are shown in Table 1.

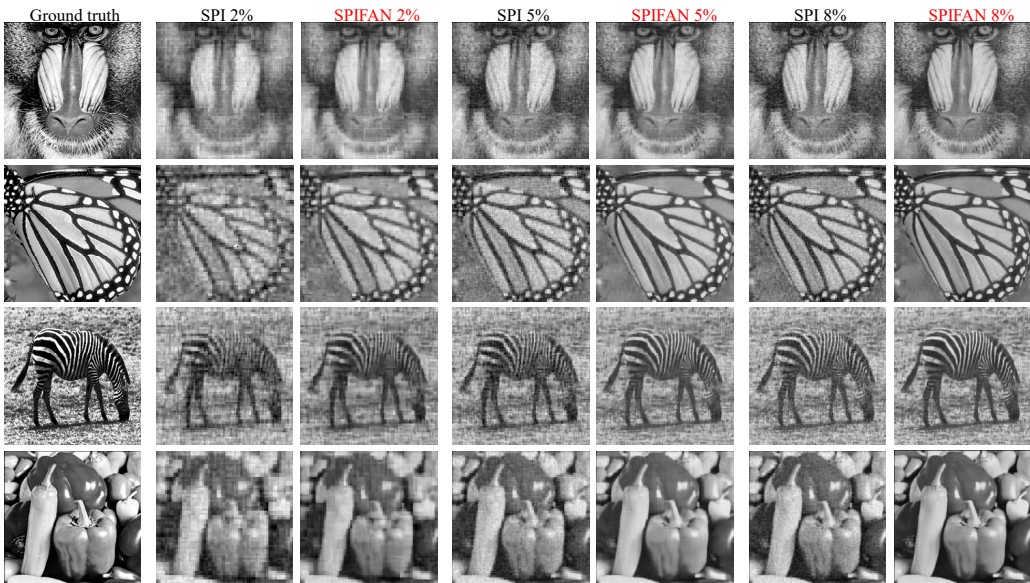

**Figure 6.** Reconstruction results of HSPI and SPIFAN at different sampling rates.

**Table 1.** Mean values of PSNRs and SSIMs for HSPI reconstructed images and SPIFAN reconstructed images at different sampling rates.

| Sample Rate | HSPI PSNR | HSPI SSIM | SPIFAN PSNR | SPIFAN SSIM |
|---|---|---|---|---|
| 2% | 12.6 | 0.39 | 17.9 | 0.59 |
| 5% | 15.9 | 0.45 | 23.9 | 0.76 |
| 8% | 17.8 | 0.64 | 24.4 | 0.83 |

As shown in Table 1, the reconstructed images from SPIFAN are significantly better than those from HSPI at the same sampling rate. It is worth noting that the PSNR (17.9) and SSIM (0.59) for the 2% HSPI reconstruction are higher than the PSNR (15.9) and SSIM (0.45) for the 5% HSPI reconstruction. The results of the qualitative and quantitative analyses show that the quality metrics of the high sampling rate SPIFAN reconstruction are superior to those of the low sampling rate SPIFAN reconstruction as the high sampling rate signal contains more precise target scene characteristics. The proposed SPIFAN is capable of reconstructing high-quality images from undersampled signals after training on undersampled signals with varying degrees of degradation.

In order to verify the superiority of the proposed method in image reconstruction performance, the reconstruction of the proposed method is compared cross-sectionally with the reconstruction of other superior algorithms. After considering the problem of balancing the sampling rate with the image quality, we chose HSPI-5% as the reconfiguration target to verify the network performance. The networks were trained using the same dataset, and Figure 7 illustrates some of the results from the experiments.

We compared the performance differences of the proposed SPIFAN method with a variety of advanced algorithms for reconstructed images at the same sampling rate. As shown in Figure 7, DCAN reconstruction resulted in a smoother reconstructed image but few visual improvements. The reconstruction results of DeepGhost and SPI-CGAN were significantly improved. It is observed that DeepGhost's reconstructed images contain significant artifacts and that the visual effect is poor. There are no artifacts in the SPI-

CGAN reconstructed image, but the reconstruction is too smooth, which affects the detailed performance of the image. The proposed SPIFAN reconstructed the best quality image, with both the subject and background of the image recovered fully, and no artifacts were generated during the reconstruction process. The mean values of the quality metrics of the reconstructed images for each algorithm for different datasets are shown in Table 2.

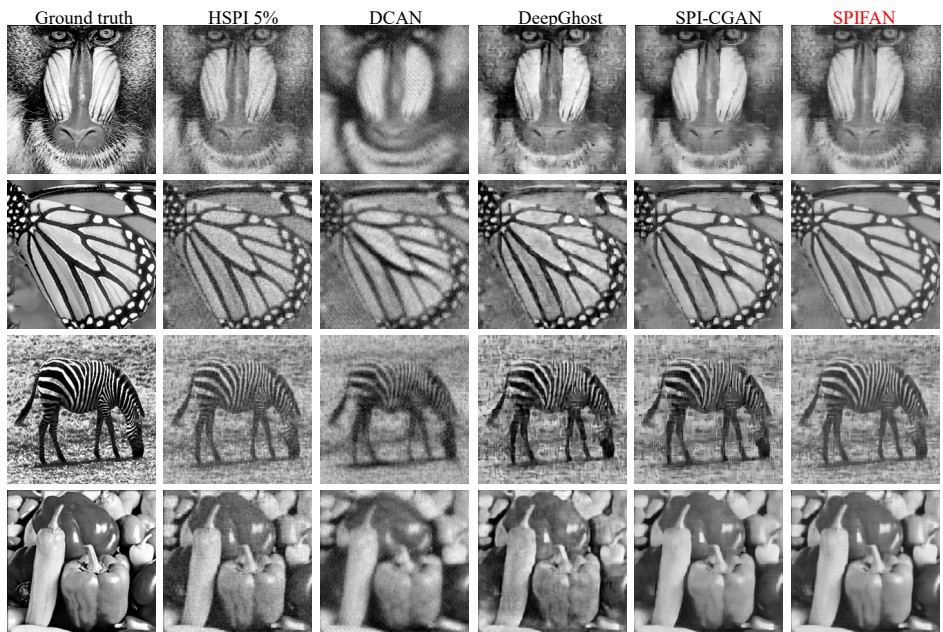

**Figure 7.** Selected results of the reconstruction of HSPI 5% images from the test set by different algorithms.

**Table 2.** Mean values of PSNRs and SSIMs of reconstruction results for the different methods when reconstructing the Set5 and Set14 datasets.

| Dataset | | HSPI 5% | DCAN | DeepGhost | SPI-CGAN | SPIFAN |
|---------|------|---------|------|-----------|----------|--------|
| Set5 | PSNR | 15.3 | 18.6 | 19.2 | 20.5 | 23.4 |
| | SSIM | 0.44 | 0.60 | 0.65 | 0.78 | 0.82 |
| Set14 | PSNR | 14.5 | 18.1 | 17.2 | 20.3 | 23.9 |
| | SSIM | 0.41 | 0.58 | 0.61 | 0.77 | 0.80 |

In Table 2, it can be seen that DCAN and SPI-CGAN have some denoising power in terms of HSPI 5% image reconstruction. DeepGhost and SPI-CGAN perform better than HSPI 5% image reconstruction. The proposed SPIFAN reconstructed images have some improvement in PSNR and SSIM compared to the results of the other reconstruction methods. It is noteworthy that SPIFAN has the least fluctuation in quality metrics when faced with image reconstruction tasks from different datasets. Therefore, it can be concluded that the reconstruction performance of SPIFAN is somewhat superior. In addition, it can be tentatively concluded that the generalization capability of SPIFAN is superior to that of other algorithms. The numerical performance of SPI-CGAN is better than that of DeepGhost reconstruction, mainly because of the negative impact of artifacts in DeepGhost reconstruction. The quality parameters of SPIFAN reconstruction are better than those of other state-of-the-art algorithms, which can prove the superiority of the proposed method in terms of reconstruction results. It is worth noting that the average quality metric of SPIFAN fluctuates the least when faced with the image reconstruction task for both datasets.

To evaluate the generalization of the proposed method, the Urban100 dataset was selected for reconstruction testing. The images in the Urban dataset are mainly architecture and landscape, which are quite different from the subjects and styles in the STL-10, Set5, and Set14 datasets, and can be used as reconstruction scenes to verify the generalization of the model. We selected the under-sampled signal with S = 5% as the reconstruction

target in order to verify the network performance. Figure 8 illustrates the results of the partial reconfiguration.

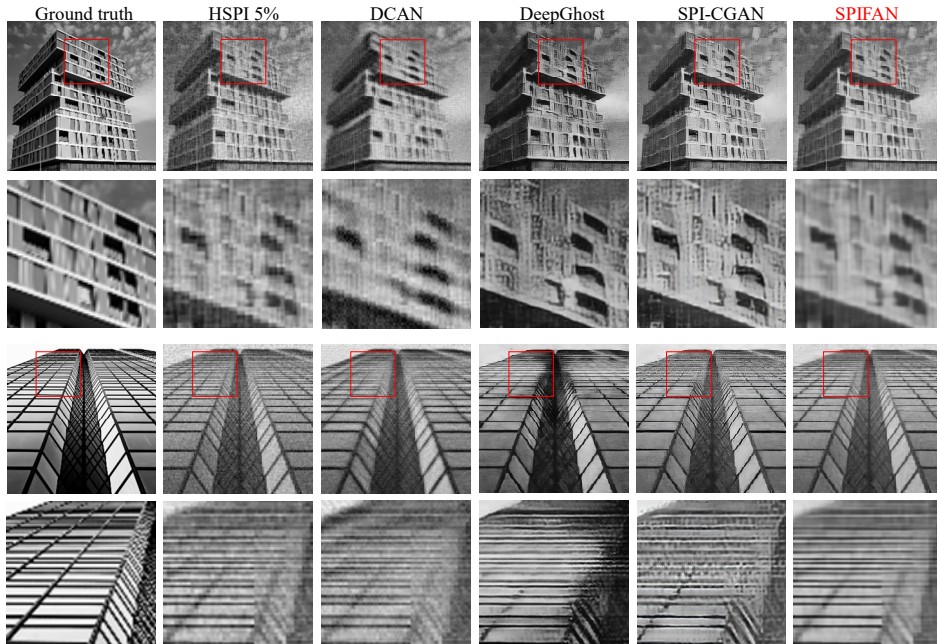

**Figure 8.** Selected results of HSPI 5% image reconstruction in Urban100 dataset by different algorithms.

Figure 8 shows the reconstruction results of each algorithm for the target scene with S = 5%, and the magnified portion allows a clearer view of the background and underlying details in the image. Clearly, only the SPIFAN reconstruction is capable of capturing the details of the image efficiently. The DCAN reconstruction is too smooth and loses image detail. The artifacts in the DeepGhost reconstruction severely damage the underlying features of the image. However, the DeepGhost reconstruction showed a significant reduction in image brightness, while the SPI-CGAN reconstruction produced a large number of artifacts. The mean values of the quality metrics of the reconstructed images for the Urban100 dataset for each algorithm are shown in Table 3.

**Table 3.** Mean values of PSNR and SSIM for the reconstruction of the Urban100 dataset by different methods.

| Dataset | | HSPI 5% | DCAN | DeepGhost | SPI-CGAN | SPIFAN |
|---------|------|---------|------|-----------|----------|--------|
| Urban100 | PSNR | 14.4 | 18.8 | 19.8 | 21.1 | 23.2 |
| | SSIM | 0.45 | 0.59 | 0.63 | 0.69 | 0.72 |

In Table 3, it can be seen that SPIFAN reconstruction is the most efficient on both quality metrics, indicating that it is also the most efficient on other types of reconstruction tasks. SPIFAN reconstruction has the smallest fluctuations (PSNR $\pm$ 0.5, SSIM $\pm$ 0.05) for different types of targets, which indicates the excellent generalization of the SPIFAN network.

### 4.3. Ablation Experiment

#### 4.3.1. Reconstruction Results

To verify the superiority of the feedback mechanism over the feedforward mechanism, we disconnected all iterations except the last, i.e., removing the high-level feature $F_H^t$ and passing it to the next iteration. Therefore, the network is unlikely to define output as the extraction of low-level features guided by high-level ones. The network is redefined as a feature-feedforward network (still retaining the recursive nature of deep learning networks), denoted SPIFAN-F (SPIFAN-Feedforward). We compared the PSNR and SSIM

values of all reconstructed images from the SPIFAN and SPIFAN-F networks following the same network training as shown in Table 4.

**Table 4.** Quality indicators for SPIFAN-F and SPIFAN reconstruction of undersampled images.

| Dataset | | SPIFAN-F | SPIFAN |
|---|---|---|---|
| Set5 | PSNR | 22.5 | 23.4 |
| | SSIM | 0.78 | 0.82 |
| Set14 | PSNR | 23.7 | 23.9 |
| | SSIM | 0.75 | 0.80 |

In Table 4, the mean PSNR and SSIM values demonstrate that SPIFAN is superior to SPIFAN-F. Therefore, it can be concluded that the feedback network is capable of producing high-quality reconstructed image predictions. Our experimental results also indicate that our proposed SPIFAN does benefit from the feedback mechanism and is not solely dependent on the recursive structure for its power.

4.3.2. Multi-Scale Attention Module

We verified the superiority of multi-scale attention modules over single-scale attention in single-pixel optimized imaging tasks. Comparative image reconstruction tests were performed using $3 \times 3$, $5 \times 5$, and $7 \times 7$ single-scale dual-attention modules, designated SPIFAN-3, SPIFAN-5, and SPIFAN-7, respectively. Combined with the data in Table 5, it can be seen that the addition of MBAM to the single-pixel imaging network has an optimizing effect on single-pixel image reconstruction, which can improve the performance of the network model and thus obtain better-reconstructed images.

**Table 5.** Quality indicators for image reconstruction using attention modules of different scales and MBAM.

| Dataset | | SPIFAN-3 | SPIFAN-5 | SPIFAN-7 | SPIFAN |
|---|---|---|---|---|---|
| Set5 | PSNR | 22.4 | 22.9 | 22.0 | 23.4 |
| | SSIM | 0.78 | 0.79 | 0.70 | 0.82 |
| Set14 | PSNR | 23.7 | 23.6 | 22.4 | 23.9 |
| | SSIM | 0.74 | 0.76 | 0.71 | 0.80 |

*4.4. Physical Experiment*

This paper is based on the SPIFAN imaging system shown in Figure 1. A 450 nm (30 W) light-emitting diode is used to illuminate a Digital Micromirror Device (DMD) (VIALUX V7000), which modulates the illumination using a pre-set Hadamard matrix. The modulation size is set to $256 \times 256$. A single-pixel photomultiplier (Hamamatsu H10493-012) converts the weak optical signal into an electrical signal to realize intensity measurement and inputs the measurement results into SPIFAN to complete the high-quality reconstruction of the undersampled image.

In physical experiments, a kettle is used as the target scene, which has a clear external outline and a complex surface pattern. This target scene can be used to test the performance of the algorithm by reconstructing it. We measured the target scene by setting S = 5% Hadamard mode and then reconstructed it using the proposed SPIFAN with other algorithms, as shown in Figure 9.

The reconstruction results in Figure 9 are similar to the results of the simulation experiments. In the DeepGhost reconstruction, there are still artifacts; the reconstructed images of DCAN and SPI-CGAN are too smooth, and their detailed features are poorly represented. Among the four optimized reconstruction algorithms, only the method proposed in this paper is able to reconstruct the external structure of the kettle and the details of the bumps on the surface excellently. In addition, the reconstructed images of each algorithm were quantitatively compared with the standard images, and the quality

metrics and imaging times of the reconstructed images of each algorithm are shown in Table 6.

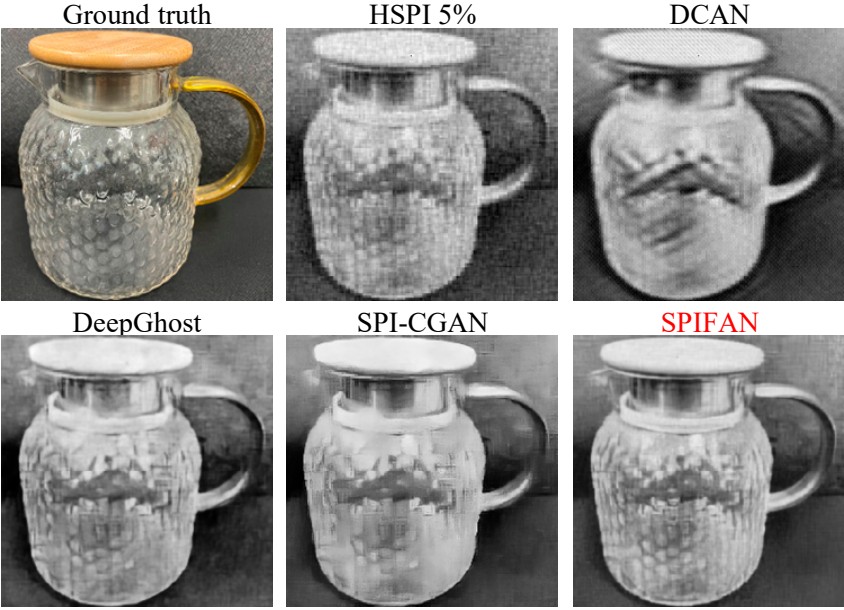

**Figure 9.** Reconstructed image of a kettle in a physical SPI system using different methods with a sampling rate of S = 5%.

**Table 6.** Quality metrics and imaging times of the kettle undersampled images reconstructed by different methods.

| Method | PSNR | SSIM | Reconstruction Time | Frame per Second (FPS) |
|--------|------|------|---------------------|------------------------|
| HSPI 5% | 13.9 | 0.37 | 54 ms | 18 |
| DCAN | 18.1 | 0.61 | 75 ms | 13 |
| DeepGhost | 18.6 | 0.66 | 102 ms | 9 |
| SPI-CGAN | 20.7 | 0.69 | 119 ms | 8 |
| SPIFAN | 22.7 | 0.79 | 88 ms | 11 |

As shown in Table 6, all four deep learning algorithms are capable of performing the reconstruction task at S = 5% in practice. In terms of quantitative performance, SPIFAN reconstruction outperforms the other advanced algorithms. The comparison of physical versus simulated experiments also reveals that the quality metrics of SPIFAN's actual reconstruction remain close to the same level, whereas other methods display significant degradation in their actual reconstruction performance. This suggests that the degradation model designed in this paper is optimized for the imaging of physical single-pixel imaging systems. As for imaging speed, SPIFAN is not the fastest but is still capable of performing real-time imaging tasks.

## 5. Conclusions

In this paper, we address the problem of compressive reconstruction of subsampled images in single-pixel imaging systems, analyze each type of degradation, and propose an attentional feedback network (SPIFAN) for single-pixel image reconstruction. The proposed network is capable of reconstructing high-quality images from undersampled signals, which is more advanced than the current state-of-the-art algorithms. We analyze the degradation patterns of single-pixel imaging systems and cover the real degradation of single-pixel imaging by assuming uniform values for the parameters of undersampling compression, Gaussian noise due to circuit variations, and Poisson noise due to the discrete nature of the charge. Using the data generated by the degradation model,

SPIFAN is trained to reconstruct undersampled images. Feedback blocks in the network use high-level representations of features to guide the extraction of low-level features under undersampling conditions, optimizing single-pixel image reconstruction by feeding back information on attentional features. In addition, Hadamard frequency domain loss is added to the course learning method, which is combined with mean loss to help the network with the single-pixel image reconstruction task. Based on experimental results, the proposed SPIFAN method shows superior results in terms of image quality metrics and visual effects as compared to traditional methods and other excellent deep learning methods. The method proposed in this paper has the potential to be improved in terms of image reconstruction rate.

**Author Contributions:** Conceptualization, Z.G.; methodology, Z.G., J.S., J.Z., Z.S. and B.L.; software, J.Z.; validation, B.L.; formal analysis, J.W.; investigation, J.W.; resources, J.Z. and J.W.; data curation, J.Z.; writing—original draft preparation, J.S. and J.Z.; writing—review and editing, Z.G., J.S. and B.L.; supervision, Z.G. and Z.S.; project administration, Z.G.; funding acquisition, Z.G. All authors have read and agreed to the published version of the manuscript.

**Funding:** This research received no external funding.

**Data Availability Statement:** All image datasets appearing in and used in this paper are derived from the following publicly available datasets: STL-10, Set4, Set14, and Urban100.

**Conflicts of Interest:** The authors declare no conflict of interest.

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
