# Peer review of "Optimal Reconstruction of Single-Pixel Images through Feature Feedback Mechanism and Attention"

_electronics, doi:10.3390/electronics12183838_

Round 1
Reviewer 1 Report
A novel neural network capable of reconstructing 2D images from experimentally 1D signals is proposed by Su and co-workers in the manuscript “Optimal Reconstruction of Single-Pixel Images Through Feature Feedback Mechanism and Attention”.
The manuscript is well-written, however, it could logically be organised better. Also, I do believe that the current state of the manuscript could be improved with some minor corrections before publishing in MDPI Electronics.
The following comments are provided:
1) I suggest the authors to define Figure 1 as the illustrative or conceptual experimental figure.
2) I suggest changing the section 4.1 title as the experimental section is placed way further (i.e., section 4.4).
3) On a similar note, I would add a simple and technical experimental figure in section 4.4.
4) Last but least I suggest the authors to cite relevant papers in the field, for instance in the terahertz and x-ray frequencies and novel spintronics applications. (Refs. [1-4])
---
References
[1] Daniele Pelliccia, Alexander Rack, Mario Scheel, Valentina Cantelli, and David M. Paganin Phys. Rev. Lett. 117, 219902 (2016)
[2] Luana Olivieri, et al., ACS Photonics 2023 10 (6), 1726-1734
[3] Cecconi V, Kumar V, Pasquazi A et al. Nonlinear field-control of terahertz waves in random media for spatiotemporal focusing [version 3; peer review: 2 approved]. Open Res Europe 2023, 2:32 (https://doi.org/10.12688/openreseurope.14508.3)
[4] Chen, SC., Feng, Z., Li, J. et al. Ghost spintronic THz-emitter-array microscope. Light Sci Appl 9, 99 (2020). https://doi.org/10.1038/s41377-020-0338-4
I suggest the authors to proofread the manuscript as some minor corrections could be made.
Reviewer 2 Report
I recommend your article for publication with minimal technical correction.
My comments also apply to the formatting of the text. The magazine "Electronics" has a simple and convenient template for designing articles https://www.mdpi.com/files/word-templates/electronics-template.dot
Pay your attention to it. This will greatly simplify the work of editors and technical proofreaders and, of course, decorate your article.
Notes:
1. Figure caption 1. Should be “Figure 1. Diagram of the experimental setup of the SPIFAN system.”
2. Line 130. Parentheses in roman type {}.
3. All variables in the article, including formulas, figures and tables, should be written in italics. Pay attention, for example, to lines 132-135, 228-236, etc.
4. There must be a space before the opening bracket, see line 132. There must also be a space between the closing bracket and the word, see figure 2 caption.
5. Figure 2, put a dot after the title of the figure. Some fragments of the picture have a too saturated background, so the inscriptions are difficult to distinguish.
6. In lines 206-209, 228-232 and beyond, check the spelling of lowercase formulas. Maybe type them in MathType, for example
?
7. Figure 3, a dot is missing after the title of the figure. Weak color contrast of the inscriptions of some fragments of the picture.
8. Maybe formula (5) Y=X+MBAM(MDC(X)).
9. In formula (6) disappeared “=”.
10. In lines 307-311, 321-323, check the spelling of characters (italics) and indices.
11. Style links MDPI:
Wang, S.; Xia, X.; Ye, L.; Yang, B. Automatic Detection and Classification of Steel Surface Defect Using Deep Convolutional Neural Networks. Metals 2021, 11, 388. https://doi.org/10.3390/met11030388
12. In the text of the article and in the figures, the preferred format for writing numbers of the form 5.1´10-3.
13. Figure 4, missing dot after title. The figure should not have line numbers.
14. Figure 5, missing dot after title. It is preferable to group fragments of the picture into a 3´3 matrix.
|
HSPI 1% |
HSPI 3% |
HSPI 5% |
|
|
|
|
|
HSPI 10% |
HSPI 15% |
HSPI 25% |
|
|
|
|
|
HSPI 35% |
HSPI 50% |
Ground truth |
|
|
|
|
|
Figure 5. Partial reconstruction of Bird images from the test set by HSPI. |
||
15. Figure 6, missing dot after title. I got the feeling that the columns of the fragments of Figure 6 with the inscription “Ground truth” duplicate each other, maybe leave one column, and enlarge the fragments of the picture?
16. Figures 7, 8 missing dot after names.
17. In table 6, spaces are missing before the “ms” dimensions.
I recommend carefully subtracting the text of the article in order to eliminate technical errors, maybe I missed something.

Reviewer 3 Report
This manuscript reports a method for completion of reconstruction of single-pixel images that incorporates a feature feedback mechanism. This manuscript is generally clearly written and appropriate comparisons are drawn between different approaches to single-pixel image reconstruction. The authors have provided an appropriate context for their work and have supported the commentary in the text with an applicable set of figures. There are some points that the authors should address and these are detailed below.
1. Page 4 line 133 “. . . and is the results of . . .” There appears to be symbol missing from this phrase. Please correct this.
2. Figure 2. The labels within this figure are largely illegible. Improve the legibility of all labelling in this figure.
3. Page 7 line 261 “Figure . Figure 3 shows . . .” There is a meaningless sentence fragment here that has to be removed.
4. Page 7 lines 285-288 “In order to . . . of this paper.” This sentence, as worded, is not very meaningful. How does design of the degradation of the image link to reconstruction of high-quality 2D images from the photomultiplier measurements? This sentence should be reworded with extra clarification given.
5. Page 8 lines 308-309 There are multiple typographical errors with repeated text that need correcting in these lines.
6. Page 15 line 547 “As shown in Table 4 . . .” It appears that the authors may mean “As shown in Table 6 . . .”
The standard of English is generally acceptable though there are some typographical errors and the manuscript would benefit from an editorial check on the grammar.
Round 2
Reviewer 1 Report
In response to all my questions and to clarify some doubts about the manuscript, I would like to express my gratitude to the authors.
